# Pharmacokinetics of Tildipirosin in Plasma, Milk, and Somatic Cells Following Intravenous, Intramuscular, and Subcutaneous Administration in Dairy Goats

**DOI:** 10.3390/pharmaceutics14040860

**Published:** 2022-04-13

**Authors:** Juan Sebastián Galecio, Pedro Marín, Verónica Hernandis, María Botía, Elisa Escudero

**Affiliations:** 1Department of Pharmacology, Faculty of Veterinary Medicine, University of Murcia, 30100 Murcia, Spain; jsgalecio@um.es (J.S.G.); pmarin@um.es (P.M.); vhb@um.es (V.H.); 2Escuela de Medicina Veterinaria, Colegio de Ciencias de la Salud, Universidad San Francisco de Quito USFQ, Diego de Robles s/n y Vía Interoceánica, Quito 170901, Ecuador; 3Interdisciplinary Laboratory of Clinical Pathology, Interlab-UMU, University of Murcia, 30100 Murcia, Spain; mbotia98@gmail.com

**Keywords:** tildipirosin, pharmacokinetics, milk, somatic cells, goats

## Abstract

Tildipirosin is a macrolide currently authorized for treating respiratory diseases in cattle and swine. The disposition kinetics of tildipirosin in plasma, milk, and somatic cells were investigated in dairy goats. Tildipirosin was administered at a single dose of 2 mg/kg by intravenous (IV) and 4 mg/kg by intramuscular (IM) and subcutaneous (SC) routes. Concentrations of tildipirosin were determined by an HPLC method with UV detection. Pharmacokinetic parameters were estimated by non-compartmental analysis. Muscle damage, cardiotoxicity, and inflammation were evaluated. After IV administration, the apparent volume of distribution in the steady state was 7.2 L/kg and clearance 0.64 L/h/kg. Plasma and milk half-lives were 6.2 and 58.3 h, respectively, indicating nine times longer persistence of tildipirosin in milk than in plasma. Moreover, if somatic cells are considered, persistence and exposure measured by the area under concentration–time curve (AUC) significantly exceeded those obtained in plasma. Similarly, longer half-lives in whole milk and somatic cells compared to plasma were observed after IM and SC administration. No adverse effects were observed. In brief, tildipirosin should be reserved for cases where other suitable antibiotics have been unsuccessful, discarding milk production of treated animals for at least 45 days or treating goats at the dry-off period.

## 1. Introduction

Bacterial infections can seriously affect livestock health, causing significant economic losses; consequently, an antibacterial intervention is critical. The development of resistant bacterial strains remains an ongoing health problem worldwide due to the frequent and inappropriate use of antibiotics. Current antibacterial agents can deal with this issue, but their application must be optimized by specific pharmacokinetic knowledge. A rational use of antibiotics must be implemented by optimizing dosing regimens in each species. Optimization can be achieved by electing a dose and schedule that result in an exposure that attains a successful clinical and therapeutic outcome. Pharmacokinetic data are often unavailable for antibiotics in some animals, principally in exotic species or small ruminants. In these circumstances, antibiotics are used under exceptional prescription by adapting or extrapolating the recommended dosage from pharmacokinetic data of other animal species. Thus, specific pharmacokinetic data are needed to adapt antibiotic therapy appropriately for particular animal species.

Contagious agalactia (CA) is a disease causing significant economic losses where small ruminants are concerned, especially those dedicated to milk production. Mycoplasma species are responsible for this disease, which displays numerous clinical signs, including mastitis, arthritis, keratoconjunctivitis, pneumonia, septicemia, and abortions, but it is evidenced differently depending on whether sheep or goats are affected. Chemotherapy, vaccines, and proper herd management procedures are the primary measures to control the disease, but clinical recurrences of contagious agalactia are common because the long-term commitment of farmers is required. The existence of asymptomatic carriers and the numerous sources of infections contribute to the accelerated disease spread inside the herd, which hinders control and prevention efforts [1]. Although these infections can be treated with several groups of antibiotics, effective treatment may be challenging because of limited therapeutic options in goats and progressive selection of resistant Mycoplasma species.

Intra-mammary infection caused by bacteria is the main cause of increased somatic cell count (SCC) in goat milk, as also in sheep and cows. The subsequent mammary gland inflammation results in a significant influx of leukocytes and, consequently, an increase in SCC in milk [2]. Moreover, in goat milk secretion, polymorphonuclear neutrophils (PMN) are the main cellular component in healthy and infected mammary glands [3,4], representing ~70% of the SSC. In this sense, it has been demonstrated that neutrophils, macrophages, and eosinophils take part in a crucial defensive role against pathogenic bacteria [5]. When predominant pathogens responsible for clinical contagious agalactia, such as *Mycoplasma agalactiae*, *M. mycoides subsp. capri*, *M. capricolum subsp. capricolum*, and *M. putrefaciens*, proliferate, they are associated with high bulk tank milk SCC in goat herds [6]. An interesting aspect is that the chemotactic factors that attract PMN to healthy mammary glands are different to those operating in glands with mastitis [7]. Macrolide antibiotics are concentrated to a large but variable extent predominately in phagocytic cells, such as polymorphonuclear leukocytes and macrophages, both in vitro [8,9] and in vivo [10]. Then, they can be an important extra-labelled option to treat CA (especially subclinical mastitis) at the dry-off period or in life-threatening cases when other antibiotics are unsuccessful. However, their use should be cautious because any detectable residues are considered illegal in Europe.

Tildipirosin is a semi-synthetic derivative of tylosin and a macrolide antibiotic currently registered for parenteral administration in cattle and pigs, especially for treating respiratory diseases induced by multiple pathogens, such as *Actinobacillus pleuropneumoniae, Mannheimia haemolytica*, *Pasteurella multocida*, and *Haemophilus parasuis*. As a macrolide, tildipirosin has proved to be effective against mycoplasma species [11]. Macrolides are bacteriostatic, although in high doses, they can be bactericidal, inhibiting essential protein biosynthesis by their selective binding to bacterial ribosomal RNA and obstructing the growth of the nascent peptide chain. Mainly, they provoke the dissociation of peptidyl-tRNA from the ribosome during the translocation process [12]. The pharmacokinetics of tildipirosin has been described in cattle, pigs, dogs, horses, sheep, and goats [13,14,15,16,17,18] where long half-lives and high bioavailability after extravascular administration have been demonstrated. Tildipirosin is promptly absorbed and rapidly distributed to tissues, being then slowly eliminated, in a characteristic macrolide fashion; low concentrations in plasma are simultaneous found with high concentrations in peripheral tissues [13,14]. High tissue concentrations as such are not usually thought to be important to efficacy, especially for extracellular organisms. However, studies using models with infected animals have demonstrated a positive correlation of efficacy with extravascular or tissue antibiotic concentrations exceeding the minimum inhibitory concentration (MIC) for infecting organisms, although plasma concentrations remained below the MIC [11,19]. Previous studies have described the pharmacokinetics of tildipirosin in goats after intravenous and subcutaneous administration at a dose of 2 and 4 mg/kg, respectively [17]. The pharmacokinetics of antibiotics may change in lactating animals. Comparative studies strongly suggest that lactation may increase the elimination rate of some macrolides from plasma. These changes in drug disposition were demonstrated for tulathromycin in goats [20] and norfloxacin in ewes [21]. However, there are no data assessed for milk disposition of tildipirosin in lactating goats.

Because of this, the purpose of this study was to characterize not only the pharmacokinetics of tildipirosin in lactating goats after intravenous (IV), intramuscular (IM), and subcutaneous (SC) administration to investigate its elimination in their milk but also the intracellular concentrations of tildipirosin reached in milk somatic cells. These parameters will provide an additional understanding of macrolides pharmacokinetics in lactating goats.

## 2. Materials and Methods

### 2.1. Animals and Treatments

Six clinically healthy Murciano–Granadina goats were selected from the Teaching Veterinary Farm at the University of Murcia (Spain). The group of goats was composed of six milking females weighing 45.8 ± 2.9 kg and aged from 3 to 5 years. The health status of the animals was determined through physical examination, hematology, clinical biochemistry (albumin, bilirubin, GGT, AST, creatinine, and urea), and a California Mastitis test. The general physical status of the goats was evaluated before tildipirosin injection and at different post-injection times (2, 12, 24, 48, and 72 h). The animals were fed with alfalfa hay and pellets free of any drug, and water was provided ad libitum. No drug was administrated for at least 30 days before or during the study. The experimental protocol was approved (CEEA 558/2019) by the Bioethical Committee of the University of Murcia (Spain).

A randomized cross-over model was designed in three periods, with a washout of 60 days between periods. Each goat received a single-dose injection of tildipirosin (Zuprevo 180, MSD Salud Animal, Salamanca, Spain) by IV (2 mg/kg), IM (4 mg/kg), and SC (4 mg/kg) administration. Intravenous injections were directly administered as a single bolus by slow injection into the left jugular vein. Subcutaneous injection was administered in the thoracolumbar region lateral to the mid-line, and IM administration was applied into the semimembranosus muscle.

Each animal was inspected daily for evidence of inflammation or discomfort signs at IM, SC, or IV injection sites by evaluating skin temperature changes; pain based on palpation at the injection sites; or swelling of the jugular, loin, or leg area. Furthermore, any clinical signs of lameness, changes in skin temperature, and swelling were recorded. Creatine kinase (CK) and haptoglobin (Hp) were determined to assess muscular damage and inflammatory response, respectively. Creatine kinase cardiac isoenzyme (CK-MB) and troponin (Tn) were measured to evaluate cardiotoxicity.

### 2.2. Sampling Procedures

Blood samples to determine the plasma concentration of tildipirosin were collected into heparinized tubes by venipuncture of the right jugular vein. Blood samples were obtained before and at 0.083, 0.167, 0.25, 0.5, 0.75, 1, 1.5, 2, 4, 6, 8, 10, 12, 24, 32, 48, 72, 96, and 120 h after tildipirosin administration. Blood samples were centrifuged at 1500× *g* for 10 min immediately after collection, and plasma was separated and transferred into duplicate Eppendorf tubes. Plasma samples were frozen at −40 °C until assayed.

Total milk production was obtained by manual milking until the mammary gland was empty. Milk samples were obtained before and at 2, 6, 12, 24, 48, 96, 144, 192, 240, 288, 336, 384, 432, 504, 576, 672, and 744 h after drug administration. The milk was collected in a container in which the volume was recorded (150 mL–2000 mL). Milk samples to determine somatic cell and milk concentrations of tildipirosin were obtained directly from the container, transferring 100 mL of milk into Falcon tubes and 2 mL into duplicate Eppendorf tubes, respectively.

Falcon tubes with milk were immediately refrigerated (4 °C) and centrifuged at 600× *g* for 10 min. The supernatant (the fat and liquid component of the milk) was removed with a Pasteur pipette to obtain a cellular pellet. Somatic cells were resuspended in 15 mL of NaCl 0.9% solution at 4 °C, homogenized vigorously with a vortex, and centrifuged at 600× *g* for 10 min and the supernatant removed again. This procedure was repeated three times until a cellular pellet containing somatic cells without extracellular components was obtained. Somatic cell pellets were frozen at −80 °C, and Eppendorf tubes with milk were stored at −40 °C until assayed.

Additional milk samples were obtained directly from the container to determine somatic cell counts. Milk samples were obtained by transferring 40 mL of milk into Falcon tubes and adding 160 µL of bronopol (50 g/L) to prevent bacterial proliferation. Finally, these samples were refrigerated until analysis, which was always performed within 24 h of collection.

To assess muscle damage, cardiotoxicity, and inflammation, additional and independent blood samples were obtained at 0 (pre-treatment), 0.5, 1, 2, 3, and 4 days after tildipirosin administration.

### 2.3. Sample Preparation

The samples of milk or plasma were prepared according to a previous study [18]. Briefly, samples were thawed and aliquots of 450 μL of milk or plasma were spiked with 10 µL of internal standard solution (tylosin tartrate 2 × 10^5^ µg/L). After mixing, 900 µL of acetonitrile was added and the mixture homogenized in a vortex (1 min). Plasma proteins were precipitated by shaking in an ultrasonic bath at 20 °C for 5 min, followed by centrifugation at 1200× *g* for 10 min. Afterward, the supernatant was extracted and evaporated for 4 h at room temperature (20 °C) in SpeedVac Vacuum concentrators (Fisher Scientific, Madrid, Spain).

Somatic cells were determined according to a previously described method [22]. Pellets of somatic cells were thawed, resuspended in 400 µL of deionized water, vortexed vigorously, and sonicated for 10 min to ensure complete cell lysis. The resulting suspension was centrifuged at 1000× *g* for 10 min, and the supernatant obtained was spiked with 40 µL of the internal standard solution (tylosin tartrate 2 × 10^5^ µg/L). Subsequently, each sample was extracted twice, with 1000 µL of diethyl ether plus 100 µL of NaOH 1M (pH = 14), in a vortex for 1 min. This solution was centrifuged at 6000× *g* for 10 min. Finally, the total supernatant was gently recollected, transferred to other polypropylene tubes, and evaporated to dryness for 45 min at room temperature (20 °C) in a SpeedVac Vacuum concentrator.

After the evaporation, the plasma, milk, and somatic cell pellet residues were reconstituted with 75 μL of the mobile phase (0.3% formic acid and acetonitrile) and 50 μL of the reconstituted residue was injected into the HPLC system.

### 2.4. Analytical Methods

Plasma, milk, and somatic cell concentrations of tildipirosin were measured by HPLC with an ultraviolet detector. An Agilent series 1220 Infinity LC HPLC system (Agilent Technologies Spain, Madrid, Spain) was equipped with a dual-gradient pump, a variable wavelength detector, and a thermostatic column compartment, connected to a Gilson 234 Autoinjector for HPLC systems (Gilson Incorporated, Middleton, WI, USA). The HPLC separation was performed using a reverse-phase Zorbax Eclipse XDB-C18 column, 150 × 3.0 mm, 5 μm particle size (Agilent Technologies Spain, Madrid, Spain), with a constant flow of 1.0 mL/min, and the column temperature was set at 30 °C. The UV wavelength was established at 289 nm. The mobile phase was composed of 0.3% formic acid (phase A) and acetonitrile (phase B). The gradient programed to analyze the plasma and the milk was (minute/A%:B%): 0–2/95:5, 15/70:30, 17/55:45, 18–20/95:5. However, the gradient selected to analyze the somatic cell pellet was 0/92:8, 7/50:50, 8–10/92:8.

A spectrophotometric technique was used to determine creatine kinase, and CK-MB was determined by spectrophotometry using an automated chemistry analyzer (AU400 Beckman Coulter Analyzer, Nyon, Switzerland). The troponin concentration was measured by an immunoassay kit (Immulite 1000). Finally, Hp was determined by a commercial colorimetric method (Tridelta Pahase range haptoglobin kit; Tridelta Development Ltd., Maynooth, Ireland).

### 2.5. Method Validation

The HPLC method using an ultraviolet detector was performed following the Food and Drug Administration guidelines [23]. No interference peaks from endogenous compounds in plasma, milk, and the somatic cell pellet were observed with tildipirosin and tylosin tartrate (internal standard) retention times. Quality control samples were prepared from a pool of blank milk, plasma, and somatic cell pellet spiked with tildipirosin. Plasma and milk were spiked with different concentrations of tildipirosin (100, 300, 600, 1000, 1666, 2500, and 3000 µg/L) and stored at −40 °C until analysis. The somatic cell pellet was spiked with different amounts of tildipirosin (0.005, 0.01, 0.02, 0.03, 0.045, 0.06, 0.08, 0.10, and 0.12 µg), incubated in a water bath at 37 °C for 60 min to promote tildipirosin uptake inside of the somatic cells, and finally stored at −80 °C until analysis. Calibrators and quality control samples were extracted as described above and injected into the chromatographic system. The linearity, the percentage of recovery, repeatability, reproducibility, the lower limit of quantification (LOQ), and the detection limit (LOD) were calculated before starting the analysis. Total tildipirosin amounts in the case of somatic cells were used for validation parameters and tildipirosin concentration in the other biological matrix.

### 2.6. Determination of Tildipirosin Concentrations in Somatic Cells

Somatic cell counts in milk were determined using a Fossomatic FC 6000 cell counter (A/S Foss Electric, Hillerød, Denmark), based on the recognition of DNA from the cells, and the total count obtained was expressed in cells/mL. Cell type distribution in milk from goats was based on previous studies [3,24] and was used for somatic cell volume calculations: macrophage (16.18%); neutrophils (63.09%); epithelial cells (13.06%); and lymphocyte (7.28%).

The concentration of tildipirosin in somatic cells (Tildi Somatic Cells) was calculated using the following relationship: Tildi Somatic Cells = (TildiPellet/VCell), where TildiPellet is the total amount of tildipirosin in the somatic cell pellet supernatant and VCell is the mean volume of somatic cells according to cell-type distribution. Here, 1.20 µL/10^6^ macrophages, 0.97 µL/10^6^ neutrophils, 1.01 µL/10^6^ epithelial cells, and 1.05 µL/10^6^ lymphocytes were used for calculations [9,25].

### 2.7. Pharmacokinetic Analysis

Pharmacokinetic parameters were estimated for each goat using the WinNonlinTM software package (WinNonlin Professional version 5.1.; Pharsight Corporation, Mountain View, CA, USA). The following non-compartmental parameters were calculated: drug concentration immediately after intravenous administration (C_0_), slowest disposition (elimination) rate constant (λ_z_), elimination half-life associated with the terminal slope (λ_z_) of a semilogarithmic concentration–time curve (t_½λz_), area under the plasma concentration–time curve from zero to infinity (AUC_0__→∞_), mean residence time (MRT), mean absorption time (MAT), systemic body clearance (Cl), apparent volume of distribution in the steady state (V_ss_), and apparent volume of distribution calculated by the area method (V_z_). AUC_0__→∞_ was calculated for any route using the linear trapezoidal rule (linear/log interpolation). Peak plasma concentrations (C_max_) and times to reach peak concentration (t_max_) were estimated directly from the experimental concentration–time curves. Bioavailability (F) was calculated by the method of corresponding areas with the following equation: F (%) = (AUC_extravascular_/AUC_intravenous_) × (Dose_intravenous_/Dose_extravascular_) × 100.

### 2.8. Statistical Analysis

Pharmacokinetic parameters were calculated for each goat and reported as the mean ± the standard deviation (SD). The harmonic mean was calculated for the half-life of elimination. The statistical analysis was performed using the IBM SPSS for Windows software package v. 26 (IBM Corporation, Armonk, NY, USA). The Shapiro–Wilk test was used to test for normality. If data were normal, a paired *t*-test was used to evaluate differences between data sets; if not, a Wilcoxon signed-rank test was used. Values were considered significantly different at *p* ≤ 0.05. Figures were plotted using ggplot2 (R version 4.0.4.).

## 3. Results

### 3.1. Analytical Method

Tildipirosin and tylosin in plasma and milk samples were eluted at approximately 4.5 and 11.0 min, respectively. The retention time for tildipirosin and tylosin tartrate in somatic cells was 3.0 min and 6.5 min, respectively. Linear regression equations; regression coefficients; repeatability; reproducibility; and recovery for plasma, milk, and somatic cells are presented in Table 1. Figure 1 shows a chromatogram of a somatic cell experimental sample where the separation quality and peak positions are visible. The repeatability and reproducibility results of this method have demonstrated reliable values for the quantitative analysis of tildipirosin in the plasma, milk, and somatic cells of goats following the established guidelines [23]. The recovery of tildipirosin from plasma, milk, and somatic cells was high (>95%). Plasma and milk LOQ and LOD were 100 µg/L and 75 µg/L, respectively. Finally, somatic cell LOQ and LOD were 0.01 µg and 0.005 µg, respectively.

Creatine kinase, Hp, CK-MB, and Tn assays showed an intra-assay and inter-assay imprecision lower than 15% and were linear after several dilutions.

### 3.2. Animals

Throughout the experiment, all goats were healthy and local or systemic adverse reactions were not observed during or after the IV, IM, and SC administration of tildipirosin. Indeed, Hp, CK-MB, and Tn showed no variations after treatment with tildipirosin. However, an increase in CK was evident after the IM administration of tildipirosin, although this rise was slightly apparent only during 24 h and with no clinical consequences. The mean values (±SD) for Hp, CK-MB, CK, and Tn concentrations in the plasma of goats after the IV, IM, and SC administration of tildipirosin are shown in Table 2.

### 3.3. Pharmacokinetic Analysis

Plasma/milk/somatic cell concentration–time curves of tildipirosin (mean ± SD) following intravenous (2 mg/kg), subcutaneous (4 mg/kg), and intramuscular (4 mg/kg) administration at single doses to goats are shown in Figure 2. Tildipirosin was detected in plasma up to 12, 48, and 72 h after intravenous, subcutaneous, and intramuscular administration, respectively. Instead, tildipirosin in whole milk was identified until day 8 (192 h) after intravenous administration, day 18 (432 h) after subcutaneous administration, and day 16 (384 h) after intramuscular injection. When somatic cell concentrations were analyzed, tildipirosin was present in any case until the last sampling day (day 18). Milk and somatic cell concentrations were parallel until 72 to 96 h, depending on the route of administration; from that moment, somatic cell concentrations remained higher than those observed in whole milk. In any case, plasma tildipirosin concentrations were less than milk and somatic cell concentrations at all sampling times.

Pharmacokinetic parameters for tildipirosin in plasma, milk, and somatic cells after the three routes of administration are presented in Table 3. C_max_ and AUC differed significantly between extravascular and intravenous administration because these parameters were not corrected by the dose. The percentages of AUC extrapolation for plasma, milk, and somatic cell data were <10.2%, <5.4%, and <8.5%, respectively.

## 4. Discussion

Tildipirosin is a newly discovered semi-synthetic 16-membered macrolide antibiotic structurally similar to tilmicosin. Reported side effects of macrolides are inflammation at the injection site, nephrotoxicity, cardiotoxicity, and hepatotoxicity. Tilmicosin can cause death in goats and other animals; therefore, this antibiotic should not be administered IV to any animal species [26,27]. The study was conducted in accordance with the principles of good clinical practice and followed a randomized, controlled, blinded, cross-over study design where no adverse effects were observed after tildipirosin administration in goats by any route. Although a single subcutaneous dose of 4 mg/kg of tildipirosin is safe for horses, cattle, and pigs [16,28], a subcutaneous dose of 4–8 mg/kg in sheep or 10 mg/kg in dogs may show cardiotoxicity [26,28]. In this study, the IV administration of tildipirosin in goats was safe in terms of cardiotoxicity. Following a single IV injection at a dose of 2 mg/kg, this macrolide did not cause cardiotoxicity since CK-MB and Tn basal concentrations did not differ significantly from those obtained several days after injection (0.25, 0.5, 1, 2, and 4 days; Table 2). Similar findings were made in ewes [18] and horses (unpublished data). However, the European Medicines Agency has reported that tildipirosin might cause restlessness, cough, and severe inflammation at the injection site [28,29]. Swelling at the injection site was not observed after IM administration of tildipirosin in goats. Nevertheless, a short-term (24 h) increase in CK was evidenced, although the magnitude of this increase was low (CK < 650 UI/L). Similar results were obtained in ewes at the same dose and via the same administration route [18].

Macrolide plasma concentrations are lower than those in tissues. This has been proved for tildipirosin in cattle [13] and pigs [14]. Consequently, they show high apparent volumes of distribution (V_z_), higher than 0.6 L/kg, corresponding to the body extracellular body fluid. In the present study, V_z_ was 7.46 L/kg, similar to the value obtained in sheep (5.36 L/kg; [18]) but much lower than that recorded in cattle, of 49.4 L/kg [13]. The half-life and MRT after IV route were 6.2 h and 14.8 h, respectively, indicating a lower persistence of tildipirosin in lactating goats than in cattle (238 h and 154 h, respectively; [13]) and sheep (17.2 h and 22.68 h, respectively; [18]).

After extravascular administration, the absorption of tildipirosin was fast and the plasma concentration reached a maximum at 1–2 h post-dose. Compared to cattle [13] and sheep [18] at the same dose, of 4 mg/kg, given subcutaneously [13], the average C_max_ was similar (0.65 µg/mL vs. 0.64 µg/mL and 0.58 µg/mL, respectively). However, after intramuscular administration (0.59 µg/L), higher values have been reported in sheep (1.26 µg/mL; [18]) and pigs (0.89 µg/mL; [14]). The mean residence time periods of tildipirosin in lactating goats (25.3 h SC and 33.1 h IM) were shorter than those obtained after subcutaneous injection in cattle (145 h; [13]) and intramuscularly in pigs (86 h; [14]). In any case, the extravascular administration of tildipirosin appeared likely to be more appropriate than intravenous injection because the plasma concentrations of the drug are sustained for an extended period.

When a comparison is established between dairy (present study) and not-dairy goats [17], persistence of tildipirosin in plasma is ephemeral (MRT_iv_ 14.8 h vs. 56.57 h and clearance 0.638 L/h/kg vs. 0.216 L/h/kg, respectively), suggesting that milk’s ion-trapping effect is an important route for tildipirosin elimination from the body.

Both absorption and general exposure to tildipirosin were ruled by dose proportionality of both C_max_ and AUC over the dose range of 2 to 4 mg/kg used in the present study, as previously reported [14,15].

Absolute bioavailability in dairy goats after subcutaneous and intramuscular administration of tildipirosin was high, indicating that this drug is rapidly and completely absorbed from the injection site. Similar high bioavailability data have been reported for goats (96.64% after SC injection [17]) but lower values in sheep (69% and 79% after SC and IM injection; [18]) and cattle (79% SC; [13]).

Tildipirosin is registered for parenteral treatment and prevention of respiratory disease in cattle and pigs. Although it can be used extra-labelled in goats, its use is banned in milking animals [28]. To date, there are no published data that report milk pharmacokinetics of tildipirosin in any species. The present study showed that tildipirosin remains in milk and somatic cells for an extended period, exceeding greatly its persistence in plasma. AUC_milk_/AUC_plasma_ and C_max-milk_/C_max-plasma_ ratios of tildipirosin greater than 50 and 55 times, respectively, are indicators of its passage into the mammary glands following systemic administration in lactating dairy goats. Tildipirosin passes readily into milk, and its concentrations were consistently higher than in plasma after all routes of administration. However, it is well known that macrolides are concentrated in PMN cells and the content of such cells in milk is high in the case of goats [3,4]. Such aforementioned high ratios are also observed in somatic cells (Table 1) and persisted longer than in plasma or whole milk, with MRT values of 141.4 ± 20.3 h and 135.1 ± 44.0 h after subcutaneous and intramuscular administration, respectively, being quantified until the last sampling time (18 days). Peak concentrations in somatic cells suppose more than 25% of tildipirosin peak concentration in whole milk, but the exposure to the antimicrobial inside somatic cells is enlarged 2 to 4 times depending on the route of administration. These high concentrations could be useful for the extra-labelled treatment of severe mastitis but also to eliminate carriers of contagious agalactia or other intracellular bacteria that cause mastitis in dairy goats, but further clinical experiments are needed to confirm this assumption. Contagious agalactia causes major economic losses and is difficult to eradicate because carriers continue to shed and infect new herds for many years after the initial infection [30]. Different statuses of immunity as well as different existing udder damage can be present in the animals. They may also be subclinically infected or may already show clinically apparent infections. Consequently, the antibiotic therapy must act quickly but long enough to cover this critical time period to avoid further spread of the disease in the herd and decrease the risk of antibiotic resistance selection [1]. As intracellular invasion by pathogens is believed to play a significant role in their systemic spread and pathogenicity. Some authors have demonstrated for Mycoplasma and S. aureus that they can be internalized and survive in a wide variety of mammalian cells, including nonprofessional phagocytes, via a mechanism that requires a specific interaction between fibronectin-binding proteins and the host cells [31,32,33].

This study demonstrated that long milk withholding periods would be necessary if tildipirosin is used in lactating goats, as is the case with tulathromycin [20]. The MRT after subcutaneous and intramuscular administration in whole milk and somatic cells was 52.7/141.4 h and 39.9/135.1 h, respectively. Other authors [34] proposed that 10 times the most prolonged terminal elimination half-life would be enough to avoid milk residues. In that regard, it could be estimated that residues of tildipirosin should not be detectable in milk after 44 days if a dose of 4 mg/kg is used in dairy adult goats after SC or IM administration. Nevertheless, as disposition kinetics may change depending on animal species and breed, it would be advisable to analyze the milk before it enters the human food chain. Additionally, farmers must comply with the extended milk withholding period if this antibiotic is used extra-labelled for treating their animals. Further studies to investigate tissue concentrations after administering single and multiple doses of tildipirosin in dairy goats are needed to determine withdrawal times accurately.

## Figures and Tables

**Figure 1 pharmaceutics-14-00860-f001:**
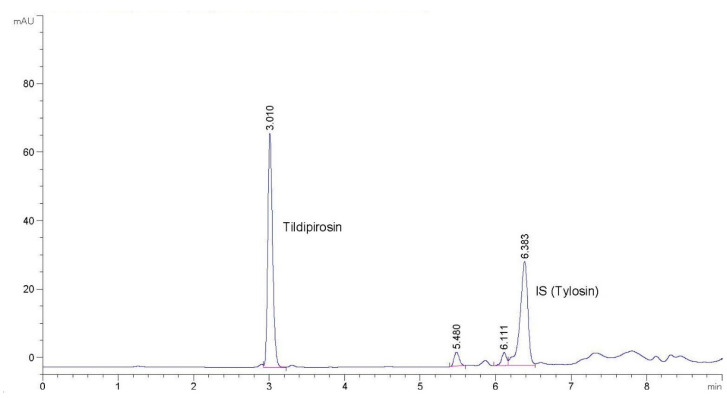
Chromatogram of tildipirosin and IS in a somatic cell experimental sample by HPLC-UV.

**Figure 2 pharmaceutics-14-00860-f002:**
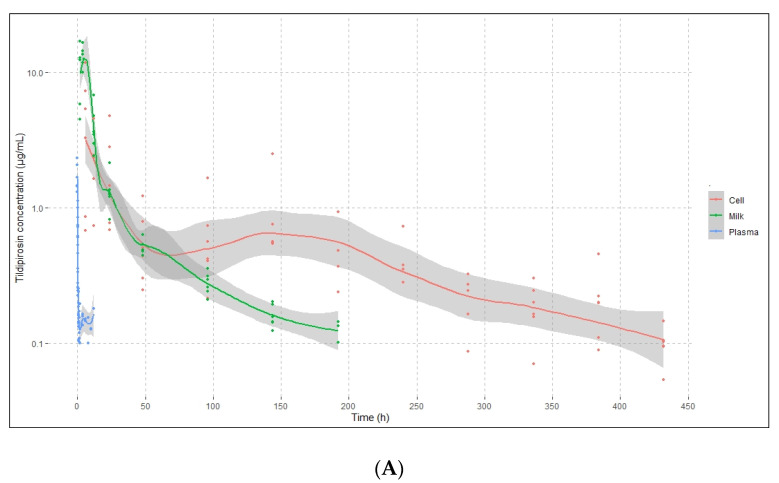
Semilogarithmic plots of intracellular somatic cell, milk, and plasma tildipirosin concentrations in goats after intravenous (**A**), subcutaneous (**B**), and intramuscular (**C**) administration. Values are the arithmetic mean ± CI 95 (*n* = 6).

**Table 1 pharmaceutics-14-00860-t001:** Validation parameters of the tildipirosin analytical method for the plasma, milk, and somatic cells of goats with the HPLC conditions of this study.

Source	Linear Regression Equation	Regression Coefficient	Repeatability (%)	Reproducibility (%)	Recovery (%)
**Plasma**	y = 0.0003x − 0.0007	R^2^ = 0.9966	2.6–6.0	10.6–19.7	95.6
**Milk**	y = 0.0003x − 0.0007	R^2^ = 0.9959	1.1–8.2	1.7–12.4	95.5
**Somatic cells**	y = 1.1365x − 0.0024	R^2^ = 0.9918	3.1–14.0	2.7–15.4	98.6

**Table 2 pharmaceutics-14-00860-t002:** Haptoglobin, creatine kinase (CK), creatine kinase myocardial band (CK-MB), and troponin concentrations (mean ± SD) in goats after subcutaneous, intramuscular, and intravenous administration of tildipirosin (*n* = 6).

		Time (Days)					
Parameter (Unit)	Route	Basal	0.5	1	2	3	4
Haptoglobin (g/L)	SC	0.92 ± 0.16	0.79 ± 0.24	0.85 ± 0.17	0.84 ± 0.26	0.80 ± 0.20	0.93 ± 0.12
Haptoglobin (g/L)	IM	0.93 ± 0.05	0.90 ± 0.12	0.82 ± 0.17	0.80 ± 0.20	0.87 ± 0.19	0.94 ± 0.11
CK (UI/L)	IM	159.4 ± 39.4	648.7 ± 119.9 ^a^	587.3 ± 200.2 ^a^	222.0 ± 89.5	159.5 ± 35.3	149.0 ± 17.8
CK MB (UI/L)	IV	137.8 ± 31.6	142.6 ± 37.8	152.5 ± 72.0	136.0 ± 20.9	143.4 ± 43.7	139.2 ± 10.1
Troponin (ng/mL)	IV	<0.05	<0.05	<0.05	<0.05	<0.05	<0.05

^a^ Indicates significant differences between basal concentrations and concentrations days after administration of tildipirosin; CK: creatine kinase; CK-MB: creatine kinase myocardial band.

**Table 3 pharmaceutics-14-00860-t003:** Pharmacokinetic parameters (mean ± SD) of tildipirosin determined individually in goats (*n* = 6) after intravenous, intramuscular, and subcutaneous administration at a single dose of 2, 4, and 4 mg/kg, respectively.

		Intravenous		Subcutaneous		Intramuscular	
Parameter	Unit	(2 mg/kg)		(4 mg/kg)		(4 mg/kg)	
**Plasma**							
C_0_	µg/mL	3.2 ± 0.9					
λ_z_	h^−1^	0.112 ± 0.104		0.081 ± 0.082		0.080 ± 0.068	
t_½λz_	h *	6.2 *		8.6 *^,a^		8.6 *^,a^	
V_Z_	L/kg	7.5 ± 2.6					
V_ss_	L/kg	7.2 ± 2.4					
Cl	L/hr/kg	0.638 ± 0.314					
AUC_0–∞_	µg·h/mL	3.7 ± 1.5		8.5 ± 3.4 ^a^		7.3 ± 3.9 ^a^	
MRT	h	14.8 ± 6.4		25.3 ± 16.0		33.1 ± 17.9 ^a^	
MAT	h			13.0 ± 11.4		20.1 ± 15.0	
C_max_	µg/mL			0.65 ± 0.23		0.58 ± 0.07	
t_max_	h			2.3 ± 1.8		1.1 ± 1.4	
F	%			118.9 ± 20.5		107.5 ± 14.9	
**Milk**							
λ_z_	h^−1^	0.012 ± 0.001		0.009 ± 0.001 ^a^		0.013 ± 0.007 ^a^	
t_½λz_	h	58.3 *		69.7 *^,a^		54.9 *^,a^	
MRT	h	36.0 ± 4.1		52.66 ± 12.11 ^a^		39.9 ± 12.9 ^a^	
AUC_milk 0–∞_	µg·h/mL	207.8 ± 29.8		475.5 ± 62.4 ^a^		523.3 ± 127.4 ^a^	
AUC_milk_/AUC_plasma_		63.4 ± 29.8		63.9 ± 41.3		56.7 ± 25.4	
t_max_	h	4.0 ± 2.2		5.3 ± 1.6		4.0 ± 0.0	
C_max_	µg/mL	14.2 ± 2.2		25.4 ± 5.9 ^a^		26.0 ± 5.7 ^a^	
C_max milk_/C_max plasma_		4.3 ± 0.7		41.4 ± 11.2 ^a^		44.7 ± 9.4 ^a^	
**Somatic Cells**							
λ_z_	h^−1^	0.007 ± 0.003		0.009 ± 0.005		0.007 ± 0.002	
t_½λz_	h	91.9 *		75.9 *		105.1 *	
MRT	h	160.6 ± 44.3		141.4 ± 20.3		135.1 ± 44.0	
AUC_SCC 0–∞_	µg·h/mL	260.1 ± 42.4		437.2 ± 169.8 ^a^		514.0 ± 194.7 ^a^	
AUC _SCC_/AUC_plasma_		81.3 ± 42.3		74.5 ± 48.3		75.3 ± 45.3	
t_max_	h	29.0 ± 16.5		9.0 ± 3.2		13.0 ± 8.8	
C_max_	µg/mL	5.2 ± 3.9		10.4 ± 7.6 ^a^		13.4 ± 10.3 ^a^	
C_max SCC_/C_max plasma_		2.0 ± 1.6		22.5 ± 18.3 ^a^		26.8 ± 19.6 ^a^	

C_0_: drug concentration immediately after intravenous administration; λ_z_: slowest disposition (elimination) rate constant; t_½λz_: elimination half-life associated with the terminal slope (λ_z_) of a semilogarithmic concentration–time curve; AUC_0–∞:_ area under the plasma concentration–time curve from zero to infinity; MRT: mean residence time; MAT: mean absorption time; Cl: systemic body clearance; V_ss_: apparent volume of distribution in the steady state; V_z_: apparent volume of distribution calculated by the area method; F: bioavailability; C_max_: peak concentration after extravascular administration; t_max_: time to reach peak concentration; * harmonic mean; ^a^ significantly different from IV (*p* < 0.05); ^b^ significantly different from SC (*p* < 0.05).

## Data Availability

The data presented in this study are available on request from the corresponding author.

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
