# Peer review of "Pharmacokinetics of Tildipirosin in Plasma, Milk, and Somatic Cells Following Intravenous, Intramuscular, and Subcutaneous Administration in Dairy Goats"

_pharmaceutics, 2022, doi:10.3390/pharmaceutics14040860_

Round 1
Reviewer 1 Report
The work is interesting, however I have a major concern about using the UV detector in HPLC method for analysis of very low concentrations of the assayed drugs in biological fluids. It is common to use MS detector or even fluorescence or electrochemical detectors in such assays, because UV detector is well known of its low sensitivity (can’t assay very low concentrations that are usually found in biological fluids), we usually use UV detector for assay of drug products, where concentrations are high enough to fit its limited sensitivity.
I tried to trace this carefully in the manuscript, so you write in 2.5 Method Validation:
(The linearity, percentage of recovery, repeatability, reproducibility, lower limit of quantification (LOQ), and detection limit (LOD) were calculated before starting the analysis),
then you wrote in the Results section 3.1 Analytical Method:
(Plasma and milk LOQ and LOD were 100 μg/L and 75 μg/L, respectively. Finally, somatic cells LOQ and LOD were 0.01 μg and 0.005 μg, respectively. )
As you can see the LOQ is very high for plasma and milk, which will indicate the method is not expected to be able to analyse the plasma and milk samples efficiently.
I could not do matching between the concentrations you found in biological fluids (Table 1) and the LOQ values, because you mention only the obtained recovery% (Table 1), not the real concentrations obtained.
Please give full explanation to this concern before we can proceed to acceptance.
Please indicate in details how you calculated LOD and LOQ
Additionally, the paper doesn’t show the chromatograms with retention times and interferences, so please add original ones to help us assess the efficiency of separations. Moreover, if you are presenting a new chromatographic method, so you need to do system suitability testing as well (please show the capacity factor, HETP, asymmetry factor…etc according to USP guidelines)
Author Response
Thank you for your positive comment. We have carefully checked and corrected all formatting errors throughout the manuscript. We have also addressed your specific comments one by one below.

Reviewer 2 Report
The manuscript presents interesting new data about tildipirosin pharmacokinetics after i.v., s.c. and i.m. administration in goats. Additionally, data about disposition of tildipirosin in the milk and in the somatic cells give valuable information about the expected withdrawal time. The manuscript is well written. The applied methods are suitable for achievement of the aim of the study. Although these advantages, minor changes are necessary before full acceptance of the manuscript.
Abstract:
The values of bioavailability can be added.
Introduction:
Line 39: exotic species or small ruminants….. – here minor species can be considered instead of exotic species. Minor species have economic impact as well as small ruminants. I think that they are more relevant in the sense of the introduction and application of tildipirosin in veterinary medicine.
Line 87: It is better to write “……using models with diseased animals (or infected animals)”. The sound of “diseased models” is not correct.
Material and methods:
Line 189: the chromatographic column is Zorbax and not Zorvax
Lines: 230-242: The calculation of pharmacokinetic parameters is well explained but only for i.v. administration. How AUC after s.c. or i.m. administration was calculated? Which method has been used (Linear up – log down ?). Please, specify. Explain which method has been used for milk and for cells.
Results
The authors have to clarify why the concentrations in the cells is given as microg/ml. At lines 261-262 the dimension for the concentrations in the somatic cells is given in microg. Please, use the same dimension or explain.
Figure 1: The third graph, C – it seems that the concentrations in the milk are increasing after 300 h. This was not discussed in the text.
Table 3:
Please add the % of extrapolation of AUC or at least state in the text was what this value for different matrices and route of administrations.
How can be explained equal values of AUC after i.m. and s.c. administration together with double differences in elimination t1/2. The answer to this question can start with the explanation about the method of determination of the elimination phase – best fit or range from certain hour to the end of sampling? Did the authors apply weighting factor? Please, check the original data again because there is some discrepancy between the figure 1C and the results in the table for the milk.
Move Cmax milk/Cmax plasma and AUC milk/AUC plasma immediately after the data about milk and after the data about somatic cells.
Discussion
First paragraph: why data from the current study after s.c. administration was not discussed in the term of toxicity? Please, revise.
Third paragraph: Cl cannot be compared between different species. For this purpose, extraction ratio should be calculated and it has to be compared. Extraction ratio takes into account cardiac output. It is better to compare t1/2 and MRT values.
The discussion about the disposition in the milk and suggestions for the withdrawal time in different animal species is valuable part of the manuscript.
Author Response
Many thanks to the editor and reviewers for their contribution to improving the manuscript. Paragraphs that were highlighted in yellow, have been modified. Figure 1 has been also improved.

Reviewer 3 Report
This is an overall well-written article. However, I suggest the authors to have the manuscript proofread by a native speaker, as there are some grammatical mistakes. The manuscript is suitable for publication after the following points have been addressed.
Line 75: Mannheimia not Mannhemia.
Table 3: Please explain in the legend the different parameters (C0, λZ, etc.)
Author Response

(The authors gave the same response as above.)

Round 2
Reviewer 1 Report
Thanks for your reply. I would recommend adding just one chromatogram showing peaks of the main analysed drugs to let the reader have an idea about separation quality and peak positions. Otherwise, we can accept it for publication.
Author Response
Thanks for your answer and also for your help improving the manuscript.
We have added a chromatogram of somatic cell experimental sample.
Best regards